# Assessing geographical and economic inequalities in caesarean section rates between the districts of Bihar, India: a secondary analysis of the National Family Health Survey

Minal Shukla,[1] Monali Mohan,[1] Alex van Duinen,[2,3] Anita Gadgil,[1] Juul Bakker,[4] Pranav Bhushan,[5] Nobhojit Roy [6]

MS and MM contributed equally.

For numbered affiliations see end of article.

**Correspondence to**
Dr Nobhojit Roy;
nobhojit.roy@ki.se

## ABSTRACT

**Background** In Bihar, one of the most populous and poorest states in India, caesarean sections have increased over the last decade. However, an aggregated caesarean section rate at the state level may conceal inequities at the district level.

**Objectives** The primary aim of this study was to analyse the inequalities in the geographical and socioeconomic distribution of caesarean sections between the districts of Bihar. The secondary aim was to compare the contribution of free-for-service government-funded public facilities and fee-for-service private facilities to the caesarean section rate.

**Setting** Bihar, with a population in the 2011 census of approximately 104 million people, has a low GDP per capita (US$610), compared with other Indian states. The state has the highest crude birth rate (26.1 per 1000 population) in India, with one baby born every two seconds. Bihar is divided into 38 administrative districts, 101 subdivisions and 534 blocks. Each district has a district (Sadar) hospital, and six districts also have one or more medical college hospitals.

**Methods** This retrospective secondary data analysis was based on open-source national datasets from the 2015 and 2019 National Family Health Surveys, with respective sample sizes of 45 812 and 42 843 women aged 15–49 years.

**Participants** Secondary data analysis of pregnant women delivering in public and private institutions.

**Results** The caesarean section rate increased from 6.2% in 2015 to 9.7% in 2019 in Bihar. Districts with a lower proportion of poor population had higher caesarean section rates ($R^2$=0.45) among all institutional births, with 10.3% in private and 2.9% in public facilities. Access to private caesarean sections decreased ($R^2$=0.46) for districts with poorer populations.

**Conclusion** Marked inequalities exist in access to caesarean sections. The public sector needs to be strengthened to improve access to obstetric services for those who need it most.

## INTRODUCTION

In many low-income and middle-income countries (LMICs), poor access to life-saving caesarean sections is a risk factor for maternal death. The caesarean section rate serves as a

### Strengths and limitations of this study

► The strength of this study is its use of secondary data from India's largest and most reliable household-level survey data source, the National Family Health Survey (NFHS).

► A limitation of the study is that the NFHS-4 was used to determine wealth quintile data because the NFHS-5 had not yet released its wealth quintile data at the time of writing this paper.

► Evidence is mixed on the optimum caesarean section rates in the population for the maximum impact on reducing maternal mortality. The disaggregated cut-offs of 10% and 20% for adequate caesarean section rates in the population were considered practical.

proxy measure of the availability of comprehensive emergency obstetric and newborn care (CEmONC) services. A rate below 5% usually denotes a lack of access to CEmONC services. Though the 'optimum' caesarean section rate remains elusive, rates >20% are not associated with any further reduction in maternal deaths.[1–3]

In India, the national caesarean section rate is climbing and almost at par with global figures. Further, the maternal mortality ratio in India has decreased from 437 (1992–1993) to 113 (2016–2017), in a quarter of a century.[4 5] Government programmes in India have contributed to increasing institutional deliveries, but the private sector has a strong role in increasing caesarean section rates. In addition, government maternal health programmes, strong advocacy, and improved infrastructure and staffing have driven the increase in caesarean section rates in LMICs.[6]

Bihar, a relatively poor state in East India bordering Nepal, has seen an increase in

the caesarean section rate from 6.2% in 2015 to 9.7% in 2019.[7 8] As this figure approaches 10%, it may suggest that the population's need for caesarean sections is met. Given the prevalent social inequalities, an optimum aggregate caesarean section rate at the state level may conceal inequities at the district level. Aggregated statistics conceal women who need but do not receive a caesarean section due to access or financial barriers by compensating with those who receive a caesarean section not medically indicated. Therefore, a more granular analysis of caesarean section rates at the district level is needed.[9]

In this paper, we report differences in caesarean section rates between population subgroups based on socioeconomic status across district wealth quintiles in Bihar. The study aimed to assess the geographical and economic inequalities in caesarean section rates between the districts of Bihar. In addition, the contributions of free government-funded public facilities and fee-for-service private facilities to population caesarean section rates are compared.

## METHODS
We conducted a secondary analysis of publicly available data from the National Family Health Survey (NFHS), a large-scale multiround survey conducted in a representative sample of households throughout India.[7 10] The NFHS provides information on health and nutrition indicators at the subnational level. The Government of India Ministry of Health and Family Welfare selected the International Institute for Population Sciences, Mumbai, as the nodal agency for the various rounds. The Woman's Schedule covered a wide variety of topics, including women's characteristics, marriage, fertility, contraception, children's immunisations and healthcare, nutrition, reproductive health, sexual behaviour, HIV/AIDS, women's empowerment and domestic violence, but not maternal deaths.

### Data sources
For Bihar, the NFHS-5 data-collection was done in 2019–2020 from 35 834 households, which included 42 843 women aged 15–49 years.[7] The NFHS records whether a caesarean section was performed in a public or private facility. The wealth quintile data were available in the 2015–2016 NFHS-4,[8 10] and raw data for Bihar were provided by the nodal agency on request.

### Study setting
Bihar, with a population in the 2011 census of approximately 104 million people,[11] has a low Gross Domestic Product (GDP) per capita (US$610) compared with other Indian states.[9 12] In total, 88% of the population of Bihar lives in rural areas. The state has the highest crude birth rate (26.1 per 1000 population) in India, with one baby born every two seconds.[7] Bihar is divided into 38 administrative districts, 101 subdivisions and 534 blocks. Each

district has a district (Sadar) hospital, and six districts also have one or more medical college hospitals.

Bihar's government healthcare system functions on three levels. At the primary level, sub-health centres and primary health centres function as the first point of contact. These primary facilities, providing basic emergency obstetric and newborn care (BEmONC) services, are mostly staffed by nurses, general physicians and complementary medicine practitioners but have no surgical capacity. At the secondary level, 149 first referral units (FRUs) function as first-level CEmONC units where caesarean sections are performed. These FRUs include district and subdistrict hospitals (serving a population of 500000–600000) and community health centres (block-level hospitals), as per Indian Public Health Standards.[13] In addition, seven government and two autonomous medical colleges in Bihar are designated as tertiary centres, four of which are in the capital city of Patna (two government medical colleges and two autonomous medical colleges). Four medical colleges are located in North Bihar, four in Patna and only one in South Bihar. Medical colleges (though tertiary) fill a gap by also functioning as secondary health facilities because FRUs are poorly functional.

Of the mothers who delivered during the last 5 years in Bihar, 25.5% had at least four antenatal clinic visits and 89.5% were protected against neonatal tetanus.[7] Most deliveries (76.2%) take place in an institution, with 56.9% in a public facility and 19.3% in a private facility.

According to the NFHS-4,[8] each household is assigned to a wealth quintile based on its household characteristics and scores established by the principal component analysis.[14] In Bihar, 80.2% of the population belongs to the two poorest quintiles, 11.4% to the middle quintile and only 8.36% to the two richest quintiles.[8] To compare districts, we combined the proportions of people in the two poorest quintiles.

### Data analysis
Caesarean section rates and the proportion of institutional deliveries for each district were extracted from the NFHS-5, including the contributions by the public and private sectors. Since no optimum rate exists, we stratified the rates into <10%, 10%–20% and >20% to measure equity in access to caesarean sections, from possible inadequate access to overuse.[15] The NFHS-4 and NFHS-5 caesarean section rates, respectively, for 2015–2016 and 2019–2020, are presented in figure 1 to demonstrate the geographical differences in change in rates between districts. Regression analysis was performed assessing the correlation between the proportion of the population in the two poorest quintiles and caesarean section rates. Both the caesarean section rate among all institutional births and the proportion of institutional deliveries in private and public health facilities were considered using the extrapolated caesarean section rate from the NFHS-5.[7]

### Patient and public involvement
Patients and the public were not involved in the development of the study design.

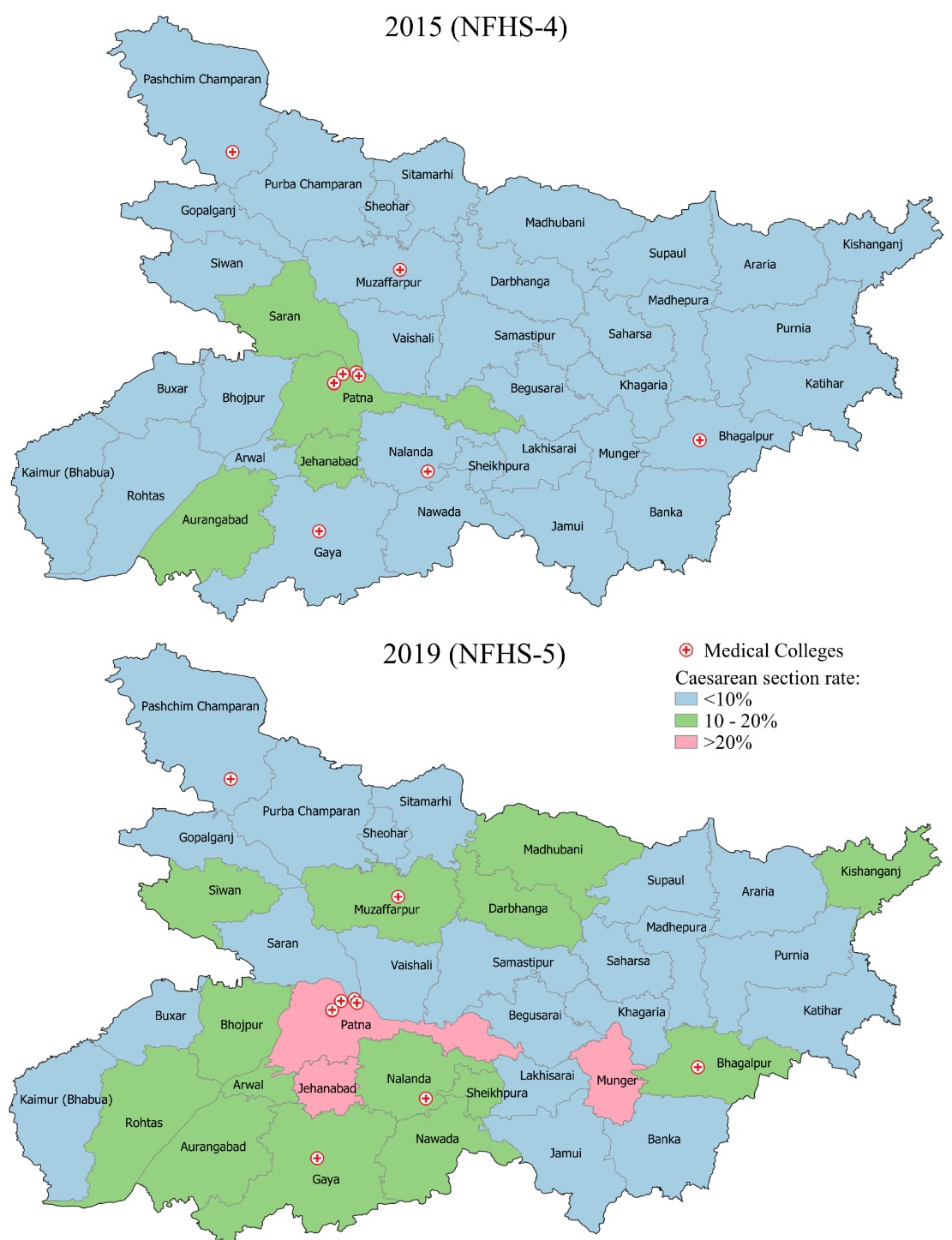

**Figure 1** Caesarean section rate by district in 2015 and 2019. Comparison of caesarean section rate in the districts of Bihar in 2015 (NFHS-4) and 2019 (NFHS-5). NFHS, National Family Health Survey.

## RESULTS

### Caesarean section rate trends across districts in Bihar

In Bihar, the caesarean section rate has increased from 6.2% in 2015 (NFHS-4) to 9.7% in 2019 (NFHS-5). Of the rate in 2019, 2.0% came from the public sector and 7.6% from the private sector. The NFHS-5 documented that the percentages of all live births delivered by caesarean section in public and private facilities were 3.6% and 39.6%, respectively.

Figure 1 shows the district-wise change in the rate of caesarean sections from 2015 to 2019. The number of districts with caesarean section rates below 10% decreased from 34 in 2015 to 21 in 2019. Using the 10% rate to evaluate access to caesarean sections in the 38

district populations, 14 districts were above 10% in 2019, three were at 10% and 21 were below 10%. Three districts (Jehanabad, Munger and Patna) had caesarean section rates above 20%.

The degree of change in caesarean section rates was uneven at the district level. Of the 38 districts, 28 demonstrated a remarkable increase of 5.9% in caesarean section rates in 4 years (from 5.5% in 2015 to 11.4% in 2019). However, against this trend was a 2.1% reduction in caesarean section rates in 10 districts (from 8.5% in 2015 to 6.5% in 2019).

Surrounding Patna are 10 adjacent districts within 86 kilometres or a 2-hour driving time. In these districts, 77.4% of the population belongs to the two poorest wealth quintiles. Moving away from the capital city, we noted a decrease in the caesarean section rate to 11.3% (of which 9.3% were in private and 2.0% were in public facilities). The more remote districts are on average 192 kilometres away from the capital, or 5–6 hours transit. Of the population in these remote districts, 81.1% are in the two poorest quintiles, and the furthest from the capital had a further caesarean section rate drop to 9.7% (of which 7.5% were in private and 2.2% were in public facilities). Public facilities conducted proportionally more caesarean sections in districts with a larger proportion of people from poorer quintiles.

Among all institutional births at public facilities, the caesarean section rate markedly decreased from 8.2% in 2015 to 2.1% in 2019. Contrary to the overall rising trend of caesarean section rate, the number of government public facilities performing at least 10% caesarean sections among all deliveries dropped from eight in 2015 to zero in 2019. The districts with university teaching hospitals and medical colleges (Bhagalpur, Nalanda, Darbhanga, Gaya, Muzaffarpur and Patna) in addition to district hospitals had district-level caesarean section rates higher than 10%, except West Champaran (6.4%).

In contrast, the caesarean section rate increased in private facilities from 6.8% in 2015 to 8.0% in 2019. Only four districts in 2015 had a caesarean section rate of >10% in private facilities, which increased to nine districts in 2019.

### Caesarean section rate trends by district wealth quintiles in Bihar

Districts with higher proportions of the two richest quintiles (Bhagalpur, Buxar, Nalanda, Patna and Rohtas) had the highest caesarean section rates among all institutional births, with 11.8% in private facilities and 2.7% in public health facilities. Patna, the capital district of Bihar (and with the wealthiest people in the state), had the highest caesarean section rate in private facilities (17.9%) and a higher than average rate in public health facilities (4.0%). Supaul is the poorest district in Bihar, with 94.1% of the population in the lowest two quintiles; of all deliveries, 1.8% and 1.9% were delivered by caesarean respectively in private and public facilities.

Figure 2 shows with decreasing wealth was a corresponding decrease in caesarean section rates (figure 2). The negative correlation ($R^2$=0.46) between population wealth quintiles and caesarean section rates in private health facilities demonstrates that access to private caesarean sections dropped dramatically in poorer populations. However, caesarean section rates dropped in poorer populations even in public facilities, though the correlation was weak ($R^2$=0.10). Public health facilities function independent of the wealth status of the population and are proportionately more available for the poor, who have lesser access to private care.

### Institutional delivery rate trends by district wealth quintiles in Bihar

The institutional delivery rate in Bihar increased from 63.8% in 2005 to 76.2% in 2019. Overall institutional births showed a negative linear correlation ($R^2$=0.43) with the wealth status of the population, and homebirths were still associated with poorer populations(figure 3). Populations in poorer districts are more reliant on public facilities for institutional deliveries and birth preparedness. Figure 3 shows that with increasing poverty, institutional deliveries in private facilities decreased, with a negative linear correlation ($R^2$=0.64). Among the poorest two quintiles, the proportion of institutional deliveries in public facilities was higher than in private facilities. In the same group, the opposite was observed for caesarean sections, which are predominantly performed in the private sector.

### DISCUSSION

In Bihar, the caesarean section rate increased from 6.2% in 2015 to 9.7% in 2019. However, caesarean sections were not equally distributed, and districts with a larger proportion of poor people had a lower rate of institutional deliveries and a lower rate of caesarean section rates compared with districts with a higher proportion of people in the two richest quintiles. The contribution of the private sector to the overall caesarean section rate was high, but it was lower in the poorer districts, while the contribution of the public sector was approximately the same.

### Caesarean sections and the health system

Caesarean section rates are driven by supply-side (such as resources within the health system, healthcare policy and strategies, health financing systems and perceptions of healthcare professionals) and demand-side (such as socioeconomic status, population preference and perceptions and trust in the health system) determinants. This study reinforces the importance of government maternal health programmes in promoting institutional deliveries in resource-poor economies. The poor population chooses to deliver in public health facilities because they are affordable and incentivised. In the poorest regions, establishing private facilities is not profitable because the paying capacity of the population is low, and therefore,

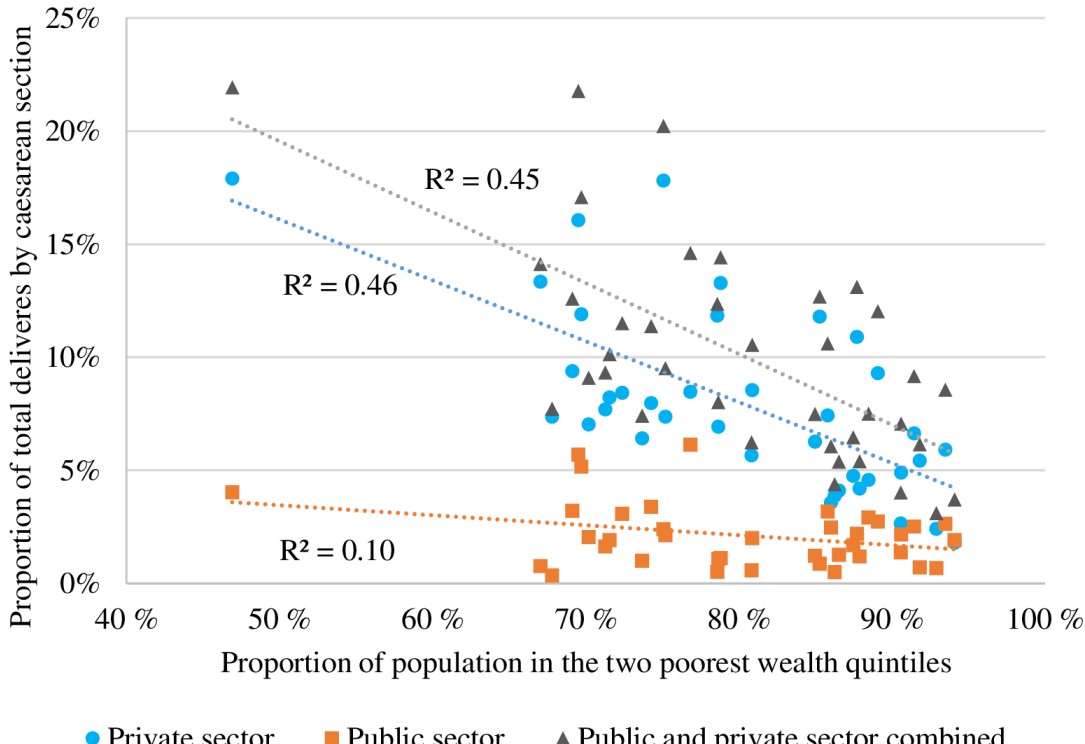

**Figure 2** Proportion of total deliveries by caesarean section in private and public sector by wealth status in the districts of Bihar in 2015 (NFHS-4) and 2019 (NFHS-5). NFHS, National Family Health Survey. The x-axis represents the proportion of the population classified in the two poorest wealth quintiles. The y-axis represents the proportion of all deliveries in the district delivered by caesarean section. For each district in Bihar, the contribution of the private sector (blue circle) and the public sector (orange square), together with the total rate of caesarean sections, are shown with individual markers. For each category, a linear regression line is added, including the coefficient of correlation ($R^2$ significant value between −1 and +1).

the private option is simply not available. The scale-up of institutional deliveries has been possible due to extensive and prolonged training programmes for auxiliary nurses and general nurse midwives, supported by task-shifted cadres of helpers in the labour delivery suite.[16]

### Uneven caesarean section rates across geographical distribution

Bihar is divided into North and South Bihar, with marked variations in geography and health.[17] North Bihar is downstream of the seasonally flooding Himalayan rivers, whereas South Bihar is in the drought zone. In both areas, the availability and utilisation of health service facilities are driven by the socioeconomic status of the population and provider factors rather than physical access.[16] Geographical access to health facilities has improved in Bihar, with better roads, electricity and transportation over the past decade.[10] The eastern districts in Bihar are insecure workplaces for healthcare professionals due to political instability. Bihar has the lowest population-to-doctor ratio in the country, and the available doctors tend to choose urban areas for government service or private practice.[18]

The caesarean section rate varies across the districts of Bihar. We found that districts closer to the capital city of Patna had higher caesarean section rates. Other factors influencing the caesarean section rate were the number of active specialists, availability of private facilities and blood availability. All these factors decrease with distance from the capital. The caesarean section rates in Bihar display the entire spectrum from too little (inadequate access) to too much (overuse), all within one state. The powerful global acceptance of technological solutions and medicalising birth is set against the local traditional preference for vaginal delivery, regardless of educational status and across wealth quintiles in the Bihar population. The poorer population faces affordability and access barriers to caesarean sections.[18] Cultural beliefs and practices about vaginal birth, irrespective of wealth quintiles, make it the preferred mode of delivery over caesarean section. Performing a caesarean section is still viewed as a lack of competence of the provider to perform an unassisted or assisted vaginal delivery, but this may also simply reflect a lack of trust or fear of a surgical procedure. Research has shown that demand for caesarean sections has been relatively uncommon in Bihar[16 18–21] due to cosmetic concerns, fear of pain during childbirth and sexual functioning concerns, foregrounded by the upper wealth quintiles from other nations[2] or other Indian states.[19]

### Distribution of caesarean section rates between public and private health facilities

Free-service public facilities are the first point of access for women with high-risk pregnancies in this predominantly

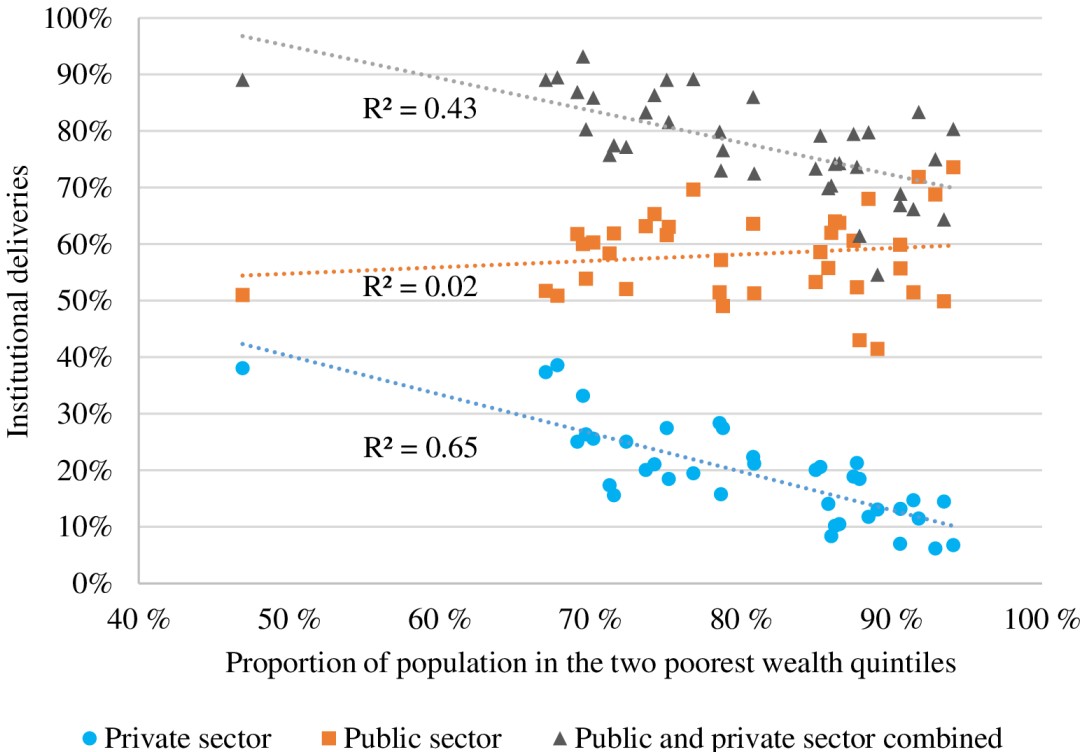

**Figure 3** Proportion of total deliveries in private and public institutions by wealth status in the districts of Bihar in 2015 (NFHS-4) and 2019 (NFHS-5). The x-axis represents the proportion of the population classified in the two poorest wealth quintiles. The yaxis represents the proportion of all deliveries in the district born in an institution. For each district in Bihar, the contribution of the private sector (blue circle) and the public sector (orange square), together with the total institutional delivery rate are shown with individual markers. For each category, a trendline and the $R^2$ result are provided. NFHS, National Family Health Survey.

poor state. However, while BEmONC services have been strengthened in public facilities, our findings show a quantitative reduction in CEmONC services available to the same women who chose to deliver in public facilities. While more than 6 out of 10 women delivered in public facilities, only around 1 in 28 women (3.6%) received a caesarean section in these public hospitals. Two in five children were delivered by caesarean section in private hospitals (39.6%). A proportionate decline occurred in the outputs and performance of public health facilities in Bihar over 5 years (2015–2019). Of 149 designated public FRUs, only 40 provide some CEmONC services, usually during office hours. The better-functioning public facilities, with significantly higher caesarean section rates, are attributed to the presence of medical colleges, teams of specialist doctors and greater urbanisation in those districts.[2] In the poorer-performing districts, low output is commonly attributed to inadequate infrastructure and staffing. However, rates remain low even in well-resourced public facilities with adequate infrastructure.[22] Our results show that caesarean section services in districts with the largest proportion of population in the poorest wealth quintiles have poor access to care in both public and private health facilities.

The supply-side provider barriers in public facilities are hesitancy to treat at-risk pregnant women, lack of clinical support for complications, administrative support for litigation and occasional episodes of violence against healthcare workers. Government doctors who refuse to perform caesarean sections in public facilities are willing to operate on the same high-risk women in private facilities. Professional social networks provide a safety net and allow for higher risk-taking in private facilities, though the infrastructure is better in public facilities. Financial incentives to providers via government schemes (Janani Shishu Suraksha Karyakram) with performance-based incentives for doctors have not increased the provision of caesarean sections at government facilities.[17] Women with high-risk pregnancies are referred to higher-level facilities (private facilities, medical colleges or tertiary facilities) for caesarean sections.[23] The demand-side patient barriers for low uptake of caesarean sections include user fees, sociocultural beliefs around the value of the procedure, a longer stay in the hospital and a trust deficit about the safety of caesarean sections. Costs in private hospitals are at least two to four times those in public hospitals.[7 24] The fee structure in the private sector is not regulated by the government, and the private sector is therefore free to decide user fees. Currently, the predominantly urban private sector in Bihar is still inadequate to provide for the needs of the 80% of the population in the two poorest wealth quintiles. Therefore public facilities need to be functional to provide free and affordable caesarean section services.

## Caesarean section distribution across wealth quintiles and equitable solutions

Caesarean section is a life-saving operation that should be available to all women. The probable reasons for the inequity and unaffordability of healthcare services include insufficient public financing and consequent out-of-pocket expenditure to meet the high costs of caesarean section services.[25]

To address inequity in access, the government of India launched a conditional cash-transfer scheme to promote institutional deliveries by providing free services to neonates and pregnant women, aiming to encourage marginalised pregnant women to use free public health facilities. More recently, the world's largest health insurance scheme, Pradhan Mantri Jan Arogya Yojana (PM-JAY), was launched in 2018 to provide free services for the 500 million poor in India. Caesarean section is included in the PM-JAY maternal package to mitigate the impact of the catastrophic expenditure incurred. However, the overall uptake of PM-JAY reimbursement rates by private providers remains poor in the state.[26]

Lessons can be learnt from other LMICs, such as Cambodia, Indonesia and Ghana, which have previously introduced policies to remove user fees for pregnant women.[2] Rwanda has set an example with its community-based health insurance scheme to reduce out-of-pocket payments for patients and a performance-based financing system to incentivise providers.[27] The key findings from our results—that although the overall caesarean rate has increased among the poor, the rates for the poorest subgroup still fall short—are also reflected by findings from Ghana and Rwanda. Like Ghana, Bihar exempts women from paying for delivery care and has a state policy of demand-side financing by incentivising institutional deliveries to reduce barriers to access to quality care in remote areas. Cambodia has identified and targeted the poor for user fee exemptions and health insurance provision through health equity funds. This scheme is similar to Indonesia's social health insurance programme, Jamkesmas, for poor and near-poor people, which has been shown to improve access and reduce financial barriers.[28] In Bhutan, only small differences in caesarean section rates between wealth quintiles have been observed, with around one in eight of all deliveries by caesarean section. However, no privatisation of healthcare exists in this LMIC.[2 29] Like Bihar, Burundi demonstrates increasing inequalities and caesarean section disparities with improved access to maternal healthcare.[30]

### Strengths and limitations

A strength of this study is its use of secondary data from India's largest and most reliable household-level survey data sources, the NFHS. It also provides insight on a more granular level into the distribution of caesarean sections on district level and its relation to economic status. Finally, the role of the public sector is an important element that was considered and essential for the understanding of the distribution of caesarean sections.

A limitation of this study is that economic status was retrieved from the NFHS-4 because the NFHS-5 had not yet released its wealth quintile data at the time of writing. In addition, no clear consensus exists on an exact optimum caesarean section rate for maximum impact on reducing maternal mortality and minimal risk for unnecessary interventions. Therefore, the cut-offs of 10% and 20% were pragmatically chosen based on the available evidence.

## CONCLUSION

Aggregated state-level caesarean section rates conceal the inequity of adequate access to emergency obstetric care among the poorest population, which is compensated by the high levels of caesarean use in the richest and most advantaged subgroups. Using relevant equity stratifiers, population subgroups and specific strategies need to be identified to address the inequities by improving access to maternal and newborn health services in public health facilities. The role of the private sector in meeting the demand for caesarean sections for the population is strongly substantiated, but the concomitant out-of-pocket expenditure is debilitating and catastrophic.

**Author affiliations**
[1]Dept of Surgery, WHO Collaboration Centre for Research in Surgical Care Delivery in LMICs, Mumbai, Maharashtra, India
[2]Faculty of Medicine and Health Sciences Department of Cancer Research and Molecular Medicine, NTNU Fakultet for ingeniorvitenskap og teknologi Trondheim, Trondheim, Norway
[3]Department of Surgery, St Olav's Hospital Universitetssykehuset i Trondheim, Trondheim, Norway
[4]Médecins Sans Frontières, Amsterdam, The Netherlands
[5]Aspirational Districts Unit, Ministry of Health and Family Welfare, Govt of India, New Delhi, India
[6]Department of Global Public Health, Karolinska Institutet, Stockholm, Sweden

**Twitter** Nobhojit Roy @#nobsroy

**Acknowledgements** We thank Priti Patil for her input in the manuscript.

**Contributors** NR, AG and AvD conceptualised and planned the research project. MS, MM, JB and PB conducted the analysis. All contributed to the writing of the manuscript. NR is responsible for the overall content (guarantor) of the manuscript.

**Funding** The authors have not declared a specific grant for this research from any funding agency in the public, commercial or not-for-profit sectors.

**Map disclaimer** The inclusion of any map (including the depiction of any boundaries therein), or of any geographic or locational reference, does not imply the expression of any opinion whatsoever on the part of BMJ concerning the legal status of any country, territory, jurisdiction or area or of its authorities. Any such expression remains solely that of the relevant source and is not endorsed by BMJ. Maps are provided without any warranty of any kind, either express or implied.

**Competing interests** None declared.

**Patient and public involvement** Patients and/or the public were not involved in the design, or conduct, or reporting, or dissemination plans of this research.

**Patient consent for publication** Not applicable.

**Ethics approval** Our study is based on the publicly available dataset of the NFHS-4 and NFHS-5 surveys with no identifiable information on the participants and can be freely accessed from the NFHS website. The ethical approval for the NFHS-4 and NFHS-5 surveys are obtained from the ethics review board of the International Institute for Population Sciences, Mumbai, India. These surveys are reviewed and approved by the ICF International Review Board. Informed written consent for participation in this survey are obtained from the respondents during

the survey. Each individual's approval is sought before the patient interview, as per the consistent methodology followed in these national surveys.

**Provenance and peer review** Not commissioned; externally peer reviewed.

**Data availability statement** Data are available in a public, open access repository. Data are available in a public, open access repository. Additionally, data can be shared by the corresponding author.

**ORCID iD**
Nobhojit Roy http://orcid.org/0000-0003-2022-7416

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
