## [Reviewer comments · BMJ Open]

ARTICLE DETAILS

TITLE (PROVISIONAL)	Assessing geographical and economic inequalities in caesarean section rates between the districts of Bihar, India: A secondary analysis of the National Family Health Survey
AUTHORS	Shukla, Minal; Mohan, Monali; van Duinen, Alex; Gadgil, Anita; Bakker, Juul; Bhushan, Pranav; Roy, Nobhojit

VERSION 1 – REVIEW

REVIEWER	Vora, K Public Health Foundation of India
REVIEW RETURNED	02-Aug-2021

GENERAL COMMENTS	Excellent paper! Needs minor revisions 1. On page 8 Line 22, there is a typo "all deliveries dropped from eight in 2015 to zero 2015." 2. On page 10, figure 2 needs some description.
--

REVIEWER	Zahroh, Rana The University of Melbourne, Centre of Health Equity, Melbourne School of Population and Global Health
REVIEW RETURNED	11-Aug-2021

GENERAL COMMENTS	Thank you for the great opportunity in reviewing this manuscript. This is a clearly written manuscript and the manuscript provides valuable findings on caesarean section inequalities in India which mirrors global phenomenon. Below are recommendations to improve the clarity and transparency of the research: Abstract 1. Page 3 Line 12-14, What contribution does this refer to? Do authors mean the contribution of funded and non-funded services to the increase/inequalities of caesarean? Please add information about this on the abstract for clarity 2. Page 3 Line 21, please add year of the surveys here Introduction 1. Page 4 line 19, I think instead >20%, it is above 9-16% threshold. Please see the study by Betran et al (2015) on the optimum caesarean section rates which is used as a basis of WHO statement on caesarean section: a. Betran, A. P., Torloni, M. R., Zhang, J., Ye, J., Mikolajczyk, R., Deneux-Tharoux, C., Oladapo, O. T., Souza, J. P., Tunçalp, Ö.,
---

Vogel, J. P., & Gülmezoglu, A. M. (2015). What is the optimal rate of caesarean section at population level? A systematic review of ecologic studies. *Reproductive health*, 12, 57.

<https://doi.org/10.1186/s12978-015-0043-6>

b. WHO statement:

https://www.who.int/reproductivehealth/publications/maternal_perinatal_health/cs-statement/en/

2. Page 5 Line 5-6, "As this figure 6 approaches 10%, it may suggest that the population's need for caesarean sections is met" I feel this statement is a bit too strong, a pragmatic threshold is 10-15%, in which anything below 10% may denote inadequate use of caesarean. So, it would be great to temper the language a bit, to acknowledge that 1) rates are not yet 10% to safely say that caesarean need is met and 2) there could be populations in Bihar which are not receiving caesarean when needed.

3. It would be great if authors can add information on rates of facility-based births (including private vs public) and antenatal care in India and Bihar overall. This information is critical for the reader to interpret the results.

Methods

1. How was the survey conducted? Seems that the survey was conducted during the pandemic (2019-2020)? I am wondering if field work was conducted or this was through online tools? And how and what type of questions asked to women in relation to caesarean section?

2. Would be great to add information on how wealth quintiles were calculated by NFHS

3. Was all caesareans experienced by a woman counted (for example 4 caesarean per woman) or only the recent birth caesarean was counted (1 caesarean per woman)? Would be great to add this information and why such decision was made

4. Please add ethical considerations by NFHS when conducting the survey (i.e. ethics approval, consent)

Results

1. Page 7 Line 14, Apologies for my ignorance, but I am wondering how did authors calculate the 2.1 2% and 7.96%, as it does not add up to 9.70% (2019 caesarean rate) or 100%? What is the numerator and denominator?

Discussion

1. On the first paragraph of the discussion, it would be great if authors can summarise the result in one paragraph before going to other discussion/implications

VERSION 1 – AUTHOR RESPONSE

Reviewer 1 (Dr. K Vora, Public Health Foundation of India)

Excellent paper! Need minor revisions

Thank you for reviewing this manuscript and your positive response.

6) On page 8 Line 22, there is a typo "all deliveries dropped from eight in 2015 to zero 2015."

Thank you for pointing this out.

Revised manuscript (Results, data; page 9, line 212: "...dropped from eight in 2015 to zero in 2019."

7) On page 10, figure 2 needs some description.

We have now improved the caption for figure 2 and 3.

Revised manuscript (results, caption figure 2; page 11, line 1): “The x-axis represents the proportion of the population classified in the two poorest wealth quintiles. The y-axis represents the proportion of all deliveries in the district delivered by caesarean section. For each district in Bihar, the contribution of the private sector (blue circle) and the public sector (orange square), together with the total rate of caesarean sections, are shown with individual markers. For each category, a trendline and the R2 result are provided.”

Revised manuscript (results, caption figure 3; page 11, line 9): “The x-axis represents the proportion of the population classified in the two poorest wealth quintiles. The y-axis represents the proportion of all deliveries in the district born in an institution. For each district in Bihar, the contribution of the private sector (blue circle) and the public sector (orange square), together with the total institutional delivery rate are shown with individual markers. For each category, a trendline and the R2 result are provided.”

Reviewer 2 (Dr. Rana Zahroh, The University of Melbourne)

Comments to the Author:

Thank you for the great opportunity in reviewing this manuscript. This is a clearly written manuscript and the manuscript provides valuable findings on caesarean section inequalities in India which mirrors global phenomenon. Below are recommendations to improve the clarity and transparency of the research:

Thank you for reviewing this manuscript and providing constructive feedback.

Abstract

8) Page 3 Line 12-14, What contribution does this refer to? Do authors mean the contribution of funded and non-funded services to the increase/inequalities of caesarean? Please add information about this on the abstract for clarity

With contribution we mean the proportion of the total caesarean section rate that is performed by the public and private sector.

Revised manuscript (Abstract; page 2, line 16): “The secondary aim was to compare the contribution of free-for-service Government funded public facilities and fee-for-service private facilities to the caesarean section rate.”

9) Page 3 Line 21, please add year of the surveys here

We have now added the year of surveys to the text.

Revised manuscript (Abstract; page 4, line 9): Further, the maternal mortality ratio in India has decreased from 437 (1992–1993) to 113 (2016–2017), in a quarter of a century. (4,5)”

Introduction

10) Page 4 line 19, I think instead >20%, it is above 9-16% threshold. Please see the study by Betran et al (2015) on the optimum caesarean section rates which is used as a basis of WHO statement on caesarean section:

a. Betran, A. P., Torloni, M. R., Zhang, J., Ye, J., Mikolajczyk, R., Deneux-Tharaux, C., Oladapo, O. T., Souza, J. P., Tunçalp, Ö., Vogel, J. P., & Gülmezoglu, A. M. (2015). What is the optimal rate of caesarean section at population level? A systematic review of ecologic studies. *Reproductive health*, 12, 57.

<https://doi.org/10.1186/s12978-015-0043-6>

b. WHO statement: https://www.who.int/reproductivehealth/publications/maternal_perinatal_health/cs-statement/en/

The optimum caesarean section rate is an important part of the study and has been widely debated. We acknowledge that there is no clear consensus here. Bertran et al. conclude that caesarean section rate

above 9-16% does not lead to improved maternal or neonatal health. However, Molina et al. (JAMA. 2015 Dec 1;314(21):2263-70) conclude that a caesarean rate of 19.1% and 19.4% were inversely correlated to maternal and neonatal mortality. Therefore, we decided to use the pragmatic benchmark of 20%.

11) Page 5 Line 5-6, "As this figure 6 approaches 10%, it may suggest that the population's need for caesarean sections is met" I feel this statement is a bit too strong, a pragmatic threshold is 10-15%, in which anything below 10% may denote inadequate use of caesarean. So, it would be great to temper the language a bit, to acknowledge that 1) rates are not yet 10% to safely say that caesarean need is met and 2) there could be populations in Bihar which are not receiving caesarean when needed. As mentioned above and now in the limitation section, there is no clear consensus on an exact optimum caesarean section rate for maximum impact on reducing maternal mortality and minimal risk for unnecessary interventions. Therefore, the cut-offs of 10% and 20% were pragmatically chosen based on the available evidence.

12) It would be great if authors can add information on rates of facility-based births (including private vs public) and antenatal care in India and Bihar overall. This information is critical for the reader to interpret the results.

We have now added information regarding antenatal care and facility-based deliveries in Bihar.

Revised manuscript (Methods, Study setting; page 6, line 23): "Of the mothers who delivered during the last five years in Bihar, 25.5% had at least four antenatal clinic visits and 89.5% were protected against neonatal tetanus. (7) Most deliveries (76.2%) take place in an institution, with 56.9% in a public facility and 19.3% in a private facility."

Methods

13) How was the survey conducted? Seems that the survey was conducted during the pandemic (2019-2020)? I am wondering if field work was conducted or this was through online tools? And how and what type of questions asked to women in relation to caesarean section?

NFHS-5 round Phase-I (For 22 States/UTs) field work was started in February 2019 and it was completed in around November 2019 (Before pandemic). The fact sheets for Phase I was published on December 12, 2020. For Phase II, the fieldwork in the 14 States/UTs was launched in January 2020. Around 38% of the fieldwork was completed in these States/UTs at the time of national lockdown which was imposed at the end of March, 2020. With the easing of lockdown, the field operations have resumed from mid-November, 2020 for Phase II. Appropriate planning, preparations and protective measures have been carried out to resume the fieldwork in Phase II States/UTs by keeping the view of the prevailing COVID-19 situation.

Pregnancy related question were asked in woman's questionnaire for children born in last five years of survey period i.e., birth occurs on or after 01 January 2014. Two questions were asked to women in relation to caesarean section specifically:

i) Was the baby delivered by caesarean section? Information was sought for last 3 births in a same questionnaire.

If more than 3 births were occurred during specified period i.e., on or after 01 January 2014 then separate woman questionnaire need to be filled as a 'continuation' sheet for the same woman.

ii) When was the decision made for the c-section, before the onset of labor or after the onset of labor?

14) Would be great to add information on how wealth quintiles were calculated by NFHS

Wealth quintile based on Household information assets expenditure and income resources relating questions divided into five groups.

Wealth quintiles are

We have now briefly explained how wealth quintiles are generated and refer to a more detailed description on the website of the DHS program.

Revised manuscript (Methods, Study setting; page 7, line 3): "According to the NFHS-4, (8) each household is assigned to a wealth quintile based on its household characteristics and scores established by the principal component analysis. (14)"

15) Was all caesareans experienced by a woman counted (for example 4 caesarean per woman) or only the recent birth caesarean was counted (1 caesarean per woman)? Would be great to add this information and why such decision was made

Information about last three caesarean sections (if experienced by a woman) were asked in a single woman questionnaire. If more than three births occurred during reporting period i.e., on or after 01 January 2014 (five years before the survey) then separate questionnaire need to be filled as a 'continuation' sheet for the same woman.

The reason for addition of only three births might be declined fertility. Normally an educated woman will go for two or three deliveries in her lifetime (India overall TFR 2.2). So, to balance urban/rural regions they have captured three births information in single form. However, provision is made to add more than three birth through filling the same woman questionnaire separately (by mentioning as a fourth or fifth birth to the last birth) as a continuation sheet for a particular woman.

16) Please add ethical considerations by NFHS when conducting the survey (i.e. ethics approval, consent)

We have now captured ethical considerations under patient and public involvement.

Revised manuscript (Methods, Patient and public involvement; page 7, line 5): "The ethical approval for the NFHS-4 and NFHS-5 was obtained from the ethics review board of the International Institute for Population Sciences, Mumbai, India. These surveys were also reviewed and approved by the ICF International Review Board. Informed written consent for participation in this survey was obtained from the respondents during the survey. Each individual's approval was sought before the interview was conducted."

Results

17) Page 7 Line 14, Apologies for my ignorance, but I am wondering how did authors calculate the 2.1 2% and 7.96%, as it does not add up to 9.70% (2019 caesarean rate) or 100%? What is the numerator and denominator?

Thank you for pointing this out, we have corrected this mistake. The calculation is as follows:

Contribution of the public sector = (proportion of all births in public facility)*(CS rate in public facility) = $0.631 \times 0.34 = 0.021 = 2.1\%$.

Contribution of the private sector: ((institutional births)-(proportion of institutional births in public facility))*(CS rate in private facility) = $(0.762 - 0.569) \times 0.396 = 0.076 = 7.6\%$

Revised manuscript (Results; page 8, line 14): "Of the rate in 2019, 2.0% came from the public sector and 7.6% from the private sector."

Discussion

18) On the first paragraph of the discussion, it would be great if authors can summarise the result in one paragraph before going to other discussion/implications

Thank you for your suggestion, we have now added a summary of the result as the first paragraph in the discussion.

Revised manuscript (Discussion; page 12, line 4): "In Bihar, the caesarean section rate increased from

6.2% in 2015 to 9.7% in 2019. However, caesarean sections were not equally distributed, and districts with a larger proportion of poor people had a lower rate of institutional deliveries and a lower rate of caesarean section rates compared to districts with a higher proportion of people in the two richest quintiles. The contribution of the private sector to the overall caesarean section rate was high, but it was lower in the poorer districts, while the contribution of the public sector was approximately the same.”

VERSION 2 – REVIEW

REVIEWER	Zahroh, Rana The University of Melbourne, Centre of Health Equity, Melbourne School of Population and Global Health
REVIEW RETURNED	06-Oct-2021
GENERAL COMMENTS	Dear authors, thank you so much for addressing the feedback that I have delivered. I have no further comments. This is a clearly written manuscript and the manuscript provides valuable findings on caesarean section inequalities in India which mirrors global phenomenon. I am happy to recommend the acceptance of this paper.